# Age-dependent changes in the anatomical and histological characteristics of the aggregated lymphoid nodules in the stomach of Dromedary camels (Camelus Dromedarius)

**Zubieda Ibrahim Hassan Omer, Jia Lu, Yu-Jiao Cheng, Pei-Xuan Li, Zhi-Hua Chen, Wen-Hui Wang**⊙*

Department of pathology, College of Veterinary Medicine, Gansu Agricultural University, Lanzhou, China

* wwh777@126.com

**Data Availability Statement:** All relevant data are within the paper.

## Abstract

Gastrointestinal associated lymphoid tissue (GALT) is an important component of the mucosal immune system. It is the largest mass of lymphoid tissues in the body and makes up more than 70% immune cells of entire body. GALT is considered to be the origin of systemic mucosal immunity and consists of solitary lymphoid nodules, aggregated lymphoid nodules (Peyer's patches, PPs), scattered lymphoid tissues, and follicular associated epithelia. PPs play important roles as antigen inductive sites of the mucosal immune system, which are mainly distributed in the intestine of animals and humans (especially ileum and appendix). However, a special area of well-developed aggregated lymphoid nodules in the abomasum of Dromedary camel was found in our laboratory. Its existence was rarely described in the stomach before. In the present study, we investigated this special structure with the dromedary camels of different ages (young, 0.5–2 years; pubertal, 3–5 years; middle-aged, 6–16 years; old, 17–20 years), by the anatomical, histological and immunohistochemical approaches. The results showed that the special structure was mainly distributed in the cardiac glandular area of the abomasum, forming a triangular area. The mucosal folds in this area were significantly thicker than those in the surrounding region. These mucosal folds had two different forms, namely reticular mucosal folds (RMF) and longitudinal mucosal folds (LMF). There were abundant lymphoid nodules in the submucosa of RMF and LMF, which were arranged in one or multiple rows. The statistical analysis of the height and thickness of RMF and LMF showed that the structure was most developed in pubertal dromedary camels. The histological characteristics of the structure were the same as PPs in the intestine of the Dromedary camel, while anatomical appearance showed some difference. The immunohistochemical examination revealed that both immunoglobulin A (IgA) and G (IgG) antibodies-producing cells (APCs) were extensively distributed in the gastric lamina propria (LP) in all age group. Our finding suggest that camel stomach not only performs digestive functions, but also involves parts of body immunity.

**Funding:** This work supported by grant sponsor: National Natural science foundation of China, grant No.31960693. Grant sponsor: National Natural Science Foundation of China; grant number: 31760723 and 30671549. Wen-Hui recieved each a ward. the funder dose not play any role in the study design, data collection and analysis, decision to publish, or preparation of the manuscript.

**Competing interests:** all The authors have declared that no competing interests exist.

## 1. Introduction

Currently, the study of mucosal immunity has greatly increased and has become one of the most active areas of immunological research. The mucosal immune system is considered as an indispensable component and the first line of defense of the systemic immune system. The coordination of the mucosal immune response relies on mucosa-associated lymphoid tissues (MALTs) which are in close contact with the mucosal surfaces in the digestive tract, respiratory tract, urogenital tract, exocrine glands, lacrimal duct, salivary glands, and breasts [1, 2]. MALTs are known to be found in all mucosal tissues [3]. The mucosal immune system operates separately from the systemic immune system, rendering MALTs significant importance in immunopathology [4]. During the body's immune response in humans and animals, the gastrointestinal tract responds effectively to invading pathogens because specialized lymphoid tissues are associated with the mucosal surfaces [5]. Gastrointestinal lymphoid tissues include lymphoid tissues associated with the gastric mucosa (gastric-MALTs) and lymphoid tissues connected with the intestinal mucosa (gut-associated lymphoid tissues, GALTs). The intestinal lymphoid tissues include Peyer's patches (PPs) in the small and large intestines, cryptopatches, mesenteric lymph nodes (MLNs), and separate lymphoid follicles scattered throughout the intestine [6–8]. PPs, solitary lymphatic follicles, and mesenteric lymph nodes (MLNs) are better known as essential sites for the induction of oral tolerance [9, 10]. GALTs are characterized by a large population of plasma cells that produce antibodies, moreover, more than 70% of the body's immune cells are found in the GALTs. The total number of plasma cells in the GALTs of the human body exceeds their number in the common spleen, lymph nodes, and bone marrow [11, 12]. The GALTs are one of the largest inductive sites in the body that trigger an immune response [13]. Furthermore, in vertebrates, the secretary immunoglobulin A (S-IgA) response is produced in the GALTs [14]. In addition to the organized mucosa-associated lymphoid nodules (such as PPs), GALTs consist of lymphocytes, dendritic cells, stromal cells, follicular dendritic cells, high endothelial venules, epithelial lymphocytes, and solitary lymphoid follicles in the lamina propria [15–19]. The organized lymphoid nodules, most commonly found in the tonsils, ileum, and appendix, may be isolated or aggregated [20]. Normally, membrane-associated lymphoid tissues are rare in the stomach of animals and exist as very small diffuse lymphoid tissues and solitary lymphoid follicles [21–23]. However, the aggregation of lymphoid nodules in the stomach mucosa has not been recorded in humans or animals including the Dromedary camel *(Camelus dromedarius)* [23–26].

Dromedary camels live in desert habitats. In order to adapt to this harsh desert environment, they have developed some unique biological characteristics and structures over a long term to withstand various surviving challenges such as changes in body temperatures, and long periods of water shortages etc. As ruminants, the stomach of camels has only three ventricles, with three distinctive glandular sac areas in the first compartment (two glandular sac areas) and second compartment (one glandular sac area). The third ventricle appears to be the abomasum. Due to the difficulty of transport to particular living environment of dromedary camels, understanding of biological and physiological properties of dromedary camels is still limited. Although there are a few reports on the mucosal immunity of the intestine, a special area of well-developed aggregated lymphoid nodules (PPs) in the stomach was not reported yet in both humans and other animals. In this study, we first described a novel PPs like structure in the stomach of Dromedary camels. Our data suggest that the stomach performs parts of immune functions.

## 2. Materials and methods

Under the laws of the Sudanese veterinary authority the camels are slaughtered for human consumption.

## 2.1. Animals

Twenty (20) healthy Dromedary camels were selected for the examination of MALTs in the third stomach compartment. The experimental camels were selected from the slaughterhouse. The camels were classified into four age groups (5 animals in each group): young (0.5–2 years old), pubertal (3–5 years old), middle-aged (6–16 years old), and old (17–20 years old). The age and sex of the camels varied in each group and the age was determined according to Schwartz's methods [26, 27].

## 2.2. Anatomical examination and measurement of the area of the special mucosal folds

The widths and heights of both longitudinal and reticular mucosal folds were measured for each age in all groups using a digital vernier caliper. The number of longitudinal mucosal folds was also counted.

## 2.3. Specimen collection

Soon after the animal was slaughtered, the stomach specimens were collected for each animal. The third department was separated from the stomach, then opened. The mucosal surface was carefully washed with normal saline. All the collected specimens were placed in 10% neutral formalin.

## 2.4. Tissue samples

Tissue samples from each animal were used to investigate the presence of mucosa-associated lymphoid tissue in the stomach. The third stomach's compartment (abomasum) of the Dromedary camel consisted of a large initial part and narrow main part. Internally, the ventral wall of the initial part had two distinct types of mucosal folds; linear and reticular mucosal folds which were different from those in the other part of the mucosa. Based on these anatomical features, the locations of the samples were determined. Tissue segments were then collected from each site of the dorsal wall, linear and reticular mucosal folds of the initial part (Fig 1). The tissue segments were placed in 10% neutral formalin.

## 2.5. Light microscopic examination and analysis

The fixed tissue samples were processed using standard histological procedures. Briefly, they were cut into blocks (1cm$^3$ each) and embedded in paraffin wax. Paraffin sections (4μm) were cut, de-waxed in dimethyl-benzene, and then processed with decreasing alcohol gradient. The samples were then treated in haematoxylin, hydrochloric acid, eosin and with increasing alcohol gradient, and finally in xylene. After processing, the specimens were mounted on glass slides with embedding medium (neutral resin) and then covered with coverslips and viewed under Olympus-DP73 optical microscope (Tokyo, Japan) for photographic images. The histological features of the abomasal tissue, including the reticular and linear parts were carefully investigated in each group.

## 2.6. Immunohistochemical procedures

The paraffin-embedded tissues of the abomasum described previously, were used for the immunohistochemistry test. Sections were made and stained with SABC-immunohistochemistry as follows: the blocks were cut (4 μm) onto a poly-lysine-coated slides. After de-waxing, antigens were retrieved with 1.0 mg/ml trypsin 1:250 (250 N.F.U./mg, Sigma USA), followed by endogenous peroxidase blocking (3% H2O2 for 15 min at 37˚C). For non-specific blocking,

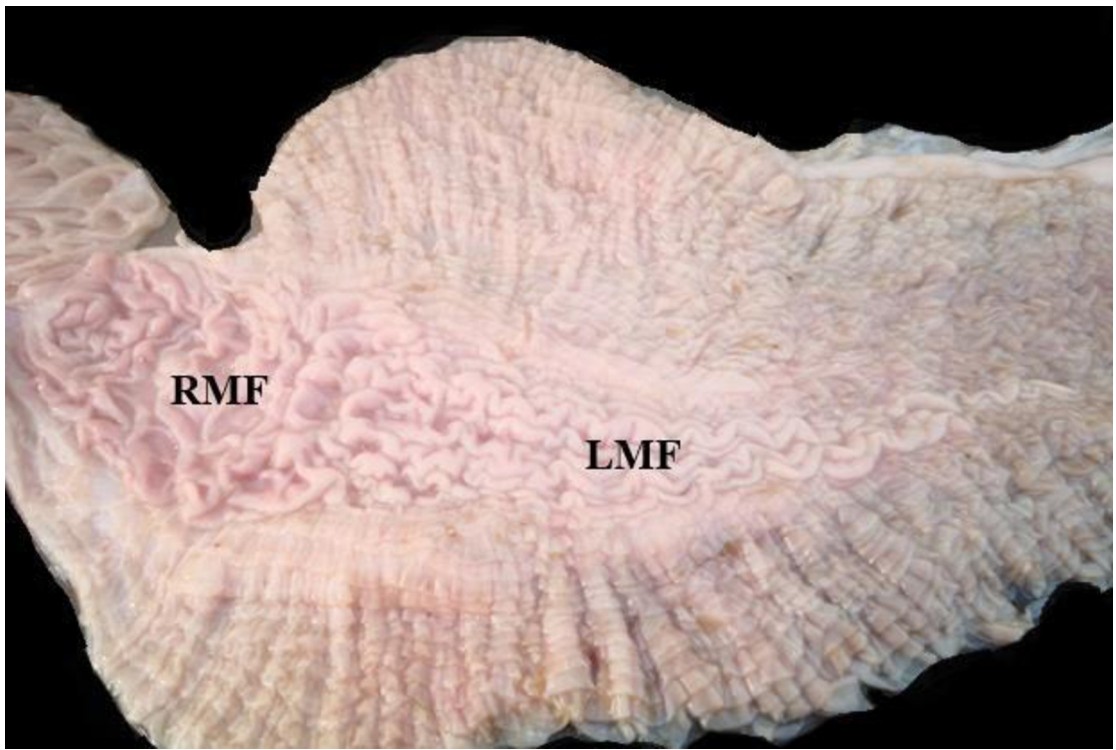

**Fig 1. Sampling locations of the cardiac glandular area of abomasum in Dromedary stomach.** RMF: reticular mucosal folds; LMF: longitudinal mucosal folds.

5% bovine serum albumin (BSA, from immunohistochemical kit, Lot No:07H3OCJ, Boster, Wuhan, Hubei, China) was used. The sections were then incubated at 4°C overnight with the primary antibody (Rabbit polyclonal antibodies against Bactrian camels IgG; synthesized in the veterinary pathology laboratory of college of veterinary medicine, Gansu agricultural university, China). Followed by (after being rinsed with PBS 3 min × 3 times) SABC-immunohistochemistry kit (Lot No.·07H3OCJ, Boster, Wuhan, Hubei, China) secondary antibody incubation for 1 h in humidified box at 37°C. After being rinsed with PBS 5 min × 4 times; the SABC was applied for 30 min in humidified box at 37°C. After being rinsed with PBS 5 min × 4 times; DAB Kit (ZSGB-BIO, Beijing, China) was used at room temperature. Specimens were stained with Hematoxylin and mounted with Neutral Balsam. Then observed, carefully investigated under Olympus-DP73 optical microscope and photographic images were taken.

### 2.7. Statistical analysis

**2.7.1. Differences in widths and heights of the mucosal folds in the area of the special mucosal folds.** Statistically significant differences in the width and height of the reticular and longitudinal mucosal folds of the abomasum ALNA between groups; were determined by one-way analysis of variance (ANOVA). Data analysis was performed using IBM SPSS V.23.0 (SPSS Inc., Chicago, USA) and ORIGINPRO was used for plotting. Differences were considered significant at $P < 0.05$.

**2.7.2. Distribution characteristics of immunoglobulin A and G in the area of the special mucosal folds.** Three sections were randomly selected from the longitudinal area. In each section, 10 microscopic fields of the LP were randomly selected, observed and

photomicrographed. The number of immunoglobulin A (IgA) and immunoglobulin G antibodies producing cells (APCs) in each microscopic field were counted, and their respective densities were calculated (Image-Pro Plus 6.0). Statistically significant differences in the distribution densities between the two APC populations in the longitudinal area of each group were determined via a one-way analysis of variance (ANOVA) followed by Duncan's multiple range test.

## 3. Results

### 3.1. Anatomical site of ALNA

The aggregated lymphoid nodule area (ALNA) was found on the ventral wall of the initial enlarged part of the abomasum and along the minor curvature. It was located in a long triangular zone between the gastric vessels and the edges of the left greater omentum attachment (Fig 2A). Opening of the abomasum along the greater curve of the stomach showed that ALNA was located posterior to the stomach flexure and extended along the wall of the minor curve of the abomasum. The region started at the origin of the initial part of the abomasum with a thickness of 5–6 cm and has reticular mucosal folds. It extended towards the abomasum flexure and along the minor curvature of the abomasum formed linear membranous folds along the minor curvature for about 16–18 cm. Hence, ALNA consisted of two unseparated regions; the reticular region with reticular membranous folds and the linear region with longitudinal membranous folds. The region converged at the abomasum flexure ending with only two folds of 1–2 cm width, forming a triangle-like zone. The mucosal folds in ALNA were dense and elevated from the surface compared to the other region which showed a distinct appearance (Fig 2B).

### 3.2. Morphological features of the mucosal surface of ALNA

In the reticular area, the mucosal folds were organized in grid-like structures. The grids were deep in young and pubertal but shallow in adult and old camels, and the bottom of the grid was flat (Fig 3). The reticular folds gradually decreased in size distally and stretched as parallel

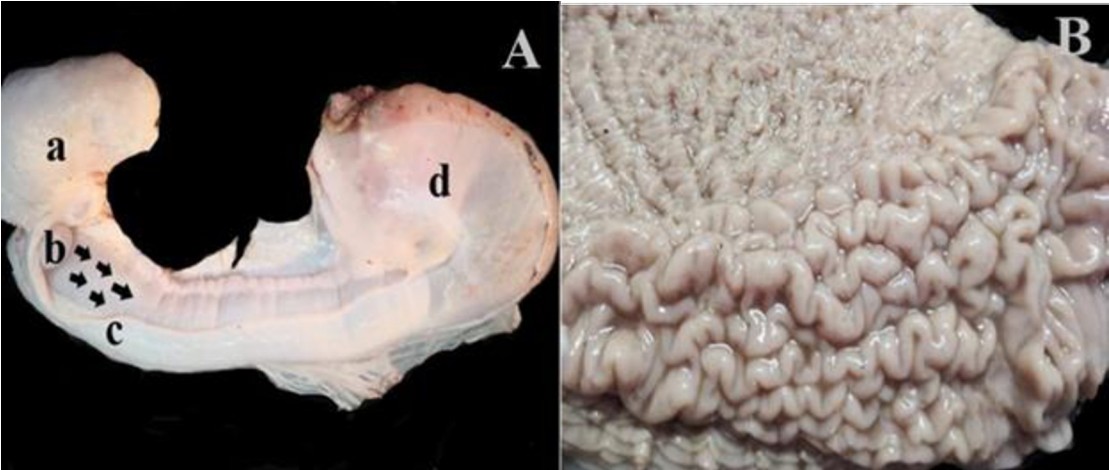

**Fig 2. The anatomical site of ALNA.** (A) Lateral view of the anatomical location of ALNA within a triangular zone (arrows); (a) the second compartment of stomach; (b) isthmus; (c) ALNA;(d) non-ALNA of the cardiac glandular area of the third compartment; (e) the fundic glandular area and pyloric glandular area of the third compartment. (B): Apparent distinction between the ALNA (lower, the mucosal folds were thick and elevated) and non-ALNA (upper, the mucosal folds were thin).

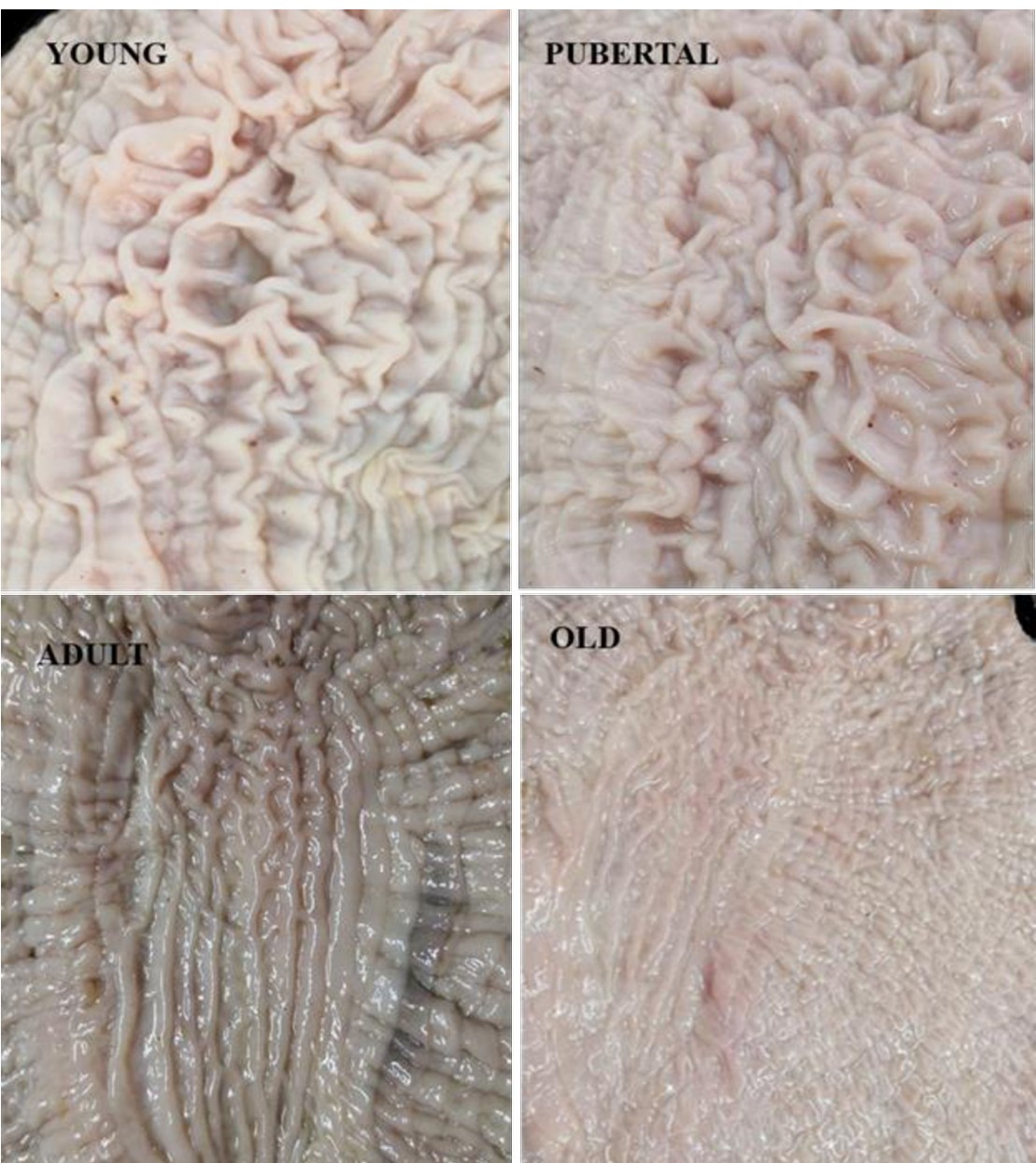

**Fig 3. The reticular area of ALNA with a grid-like appearance.** Young: 1 year old; pubertal: 5 years old; adult: 12 years old; old: 19 years old.

folds distributed along the wall of the lesser carve of the stomach forming the longitudinal region (Fig 3). The longitudinal folds were interconnected by small branchial folds.

Morphologically, ALNA was well-developed in young and pubertal animals and exhibited abundant mucosal folds, whereas the region in adult and old camels, had only a few thin mucosal folds. The size of the region gradually decreased with age but did not disappear completely in old animals (Fig 4). Overall, our data suggest that the anatomical location of the clustered lymphoid follicles was similar in all age groups, whereas the size and degree of development varied with age.

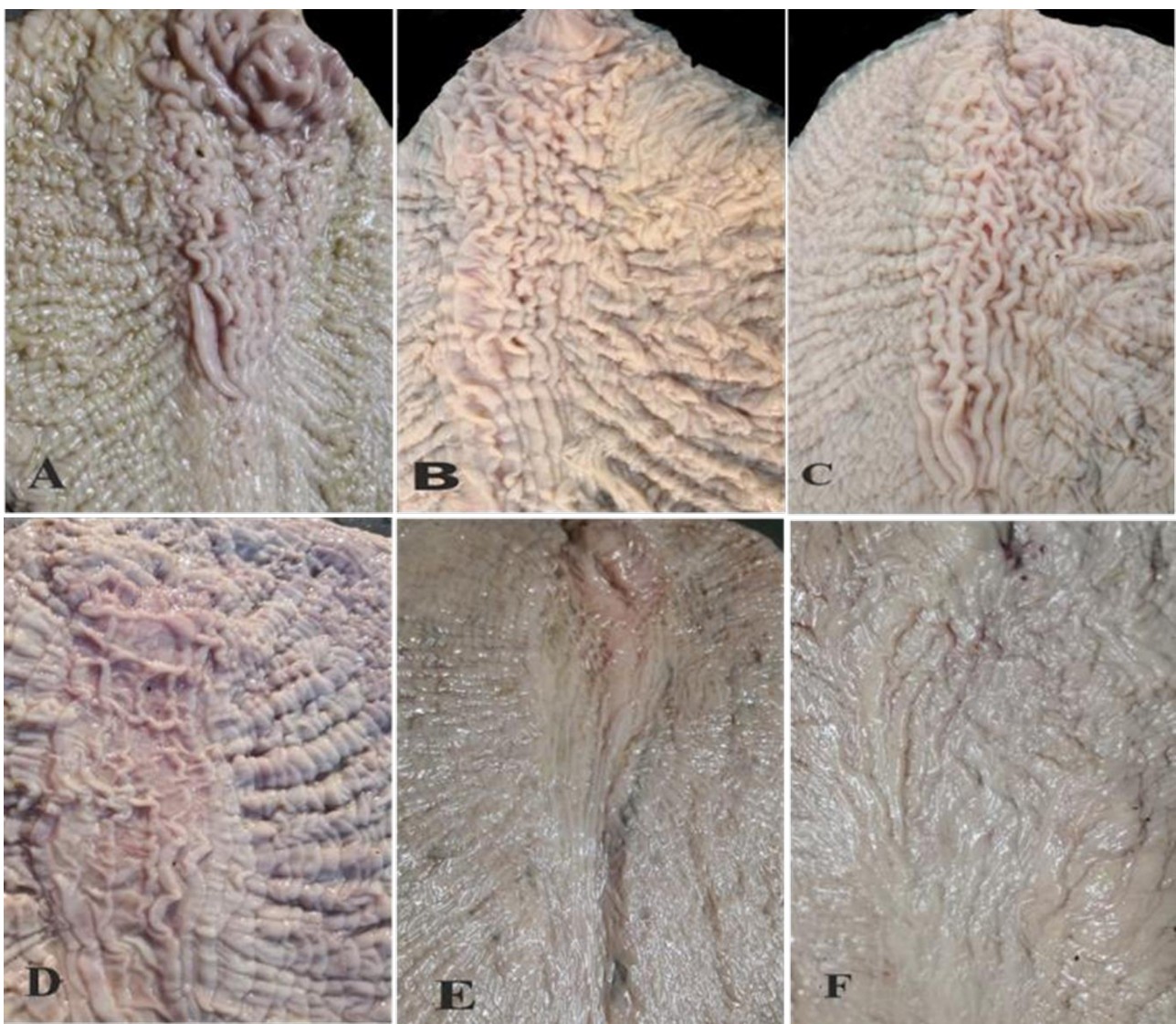

**Fig 4. Mucosal view of ALNA in Dromedary camel in different ages.** The mucosal folds were more and well-formed in young (A) and pubertal camels (B). Mucosal fold's number, height, and width decreased with age increment. A: 6months old; B: 4 years; C: 7years old; D: 15years old; E: 17year old & F: 20 years old.

### 3.3. Age differences on the mucosal surface of ALNA

Statistical analysis showed that ALNA varied with age. In all groups, the width and height of both longitudinal (Fig 5) and reticular (Fig 6) membrane folds significantly increased in young, peaked in pubertal animals, and then gradually decreased with age. There were significant differences (p < 0.05) in heights and widths of the longitudinal folds between all age groups (Fig 5), except for longitudinal folds width in 1.5 years and 10 years old animals (Fig 5). In the reticular mucosal fold area, the differences in the mucosal folds' heights were also significant between all groups, except for those in 8 months and 10 years old animals (Fig 6, p > 0.05). The reticular folds were higher but narrower than the linear folds. The membranous folds of both longitudinal and reticular regions were more developed in pubertal and young animals than adults and old animals. Accordingly, the number of these longitudinal mucosal

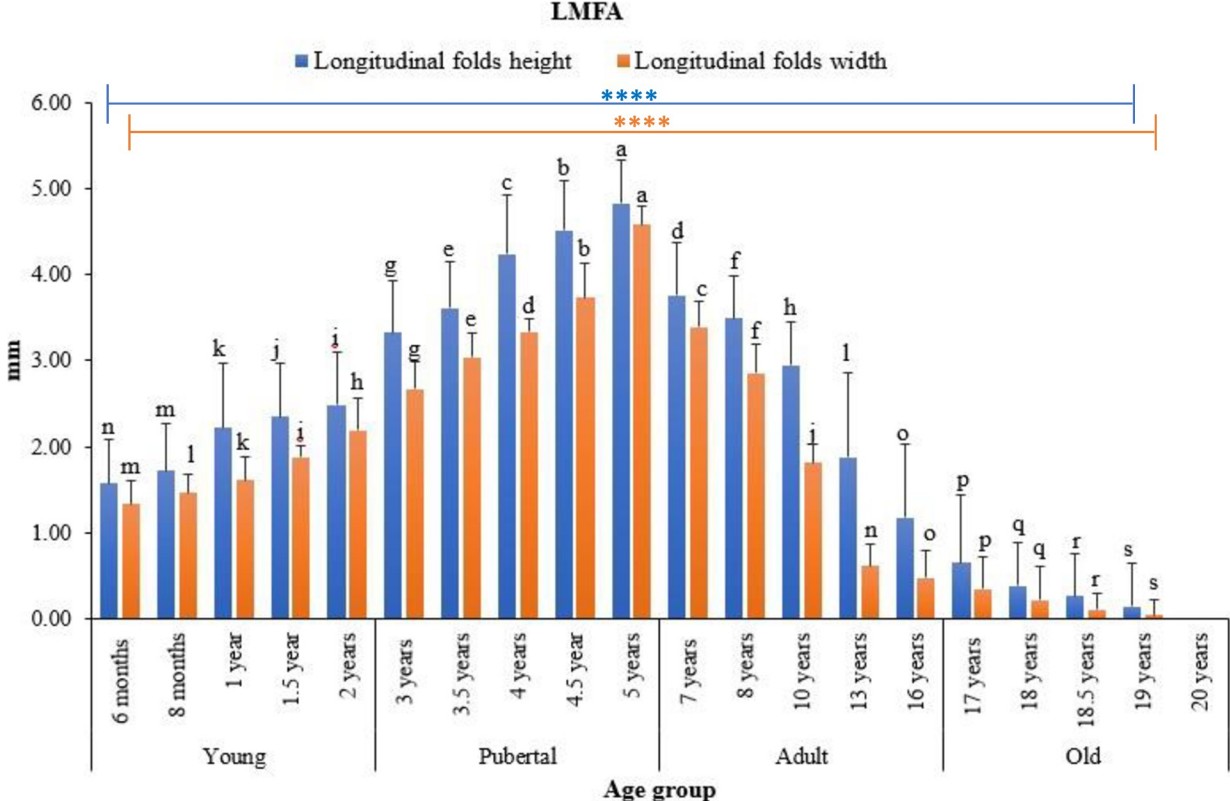

**Fig 5. Differences in height and width of the longitudinal mucosal folds of ALNA between age groups of dromedary camels.** Data are expressed as means (n = 3). Error bars represent standard deviation. Bars with a different letter are significantly different (one-way ANOVA with Duncan's multiple range test, p < 0.05). **** represents significance at p < 0.0001. LMFA: longitudinal mucosal fold area.

folds was abundant in pre-puberty camels, peaked at pubertal age, and decreased thereafter. There were no significant differences in the height and width of both types of mucosal folds in camels of the same age group.

## 3.4. Histological observation of the lymphoid tissues on the mucosa of cardiac glandular area

The histological studies discovered ALN in the mucosa of the cardiac glandular area, forming a distinct area with different morphological features. The mucosal surface of this area formed distinct membranous folds compared to the adjacent area without accumulation of lymphoid tissues. ALNA consisted of aggregated lymphoid follicles. The clustered lymphoid masses included various lymphoid cells and were primarily located in the submucosa of the stomach. Together with the mucosa they protracted towards the mucosa and formed mucosal folds. In young and pubertal camels, the clustered lymphoid follicles were broad, abundant, and well-formed compared to adults and old camels. With age, the size and number of follicles shrunk, and subsequently, the size of the mucosal folds decreased (Fig 7).

The follicles had different shapes and sizes including round, oval and irregular shapes. Basically, the follicles were distributed in one row in the submucosa along the mucosal fold (Fig 8A). However, some follicles were organized in two (Fig 8B) or more layers, especially at broad and bifurcated mucosal folds (Fig 8C).

The mucosa, mucosal gland, and isolated lymphoid tissue in the lamina propria overlayed the lymphoid follicles. The follicles were isolated by extensive connective tissue filling the

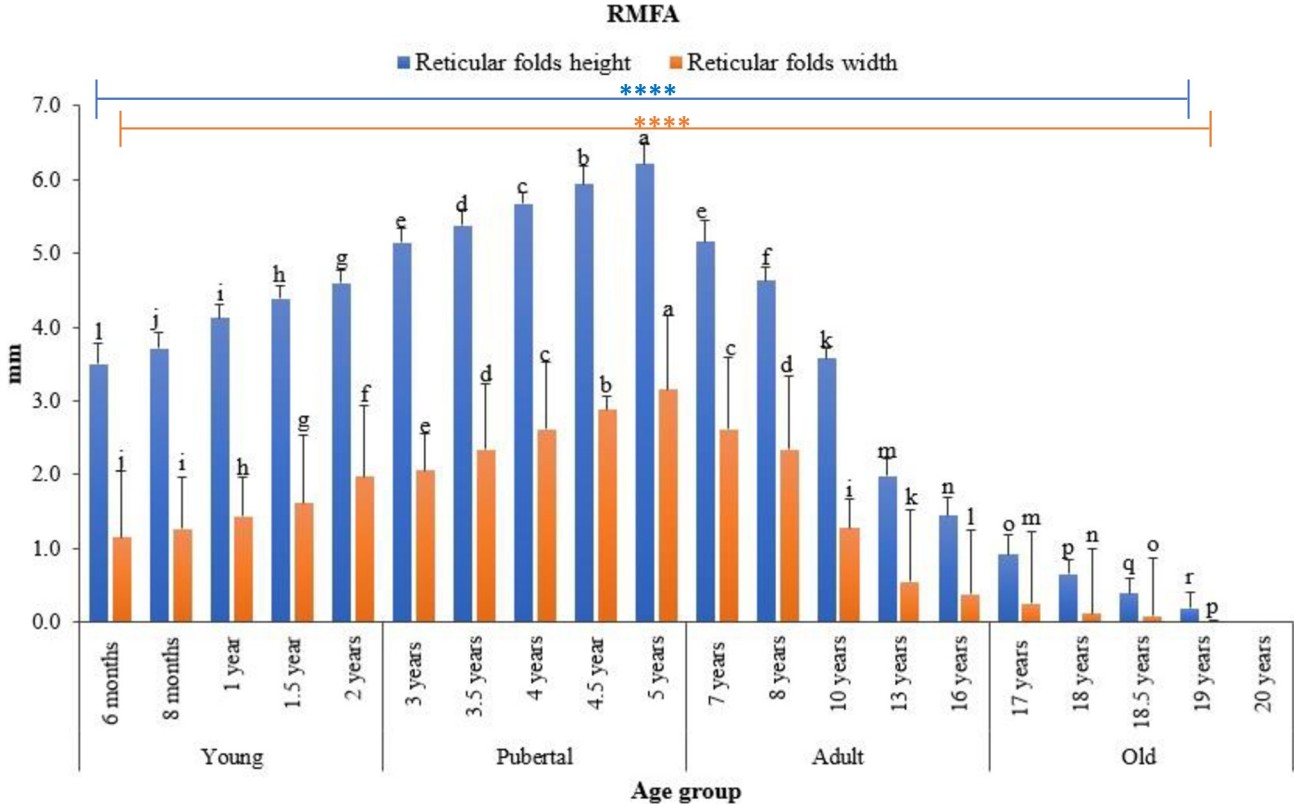

**Fig 6. Differences in height and width of the reticular mucosal folds of ALNA between age groups of dromedary camels.** Data are expressed as means.Error bars represent standard deviation. Bars with a different letter are significantly different (one-way ANOVA with Duncan's multiple range test, p < 0.05). **** represents significance at p < 0.0001. RMFA: Reticular mucosal folds area.

interfollicular areas. This extensive connective tissue contained high endothelial venules and scattered lymphoid tissues. The lymphoid tissue consisted of different types of lymphoid cells distributed on and between the fibers (Fig 9). The diffuse lymphatic tissue was also found among the mucosal glands. The clustered follicles had a distinct geminal center, which was lightly stained and contained mainly several cell types including plasma cells. In contrast, the peripheral area was intensely pigmented and contained T-cells.

Several follicles protruded into the typical upper part of the lamina propria and mucosa forming bottled-shaped follicles. These follicles were usually free of glands, coated with follicle-associated epithelium (FAE), and had a characteristic subepithelial dome area (SED). Some nodes bulged into the lamina propria at some sites but were more prominent without FAE.

### 3.5. immunohistochemical observations

The distributions of IgA (Fig 10) and IgG (Fig 11) APCs, in both the longitudinal and reticular areas of aggregated lymphoid nodules were similar in the young group. They were dispersed in the lamina propria (LP) mainly between the gastric glands. There were no antibody-producing plasma cells were observed in the smooth muscle layer nor the epithelium. In each sample of the reticular and longitudinal area from camels in the other groups, the distribution criteria of IgA APCs were similar to those in the young group, and the distributions of IgG APCs were similar to those of IgA APCs within the LP.

**Fig 7. The histological appearance of ALNA.** The mucosal folds were well developed and the lymphoid follicles were abundant in young (A) and pubertal (B) but poorly arranged with few follicles in old animals (C and D). In young and pubertal, some mucosal folds were broad and bifurcated (B). A: 7 months old; B: 4 years old; C: 17 years; D: 20 years. H and E (40x).

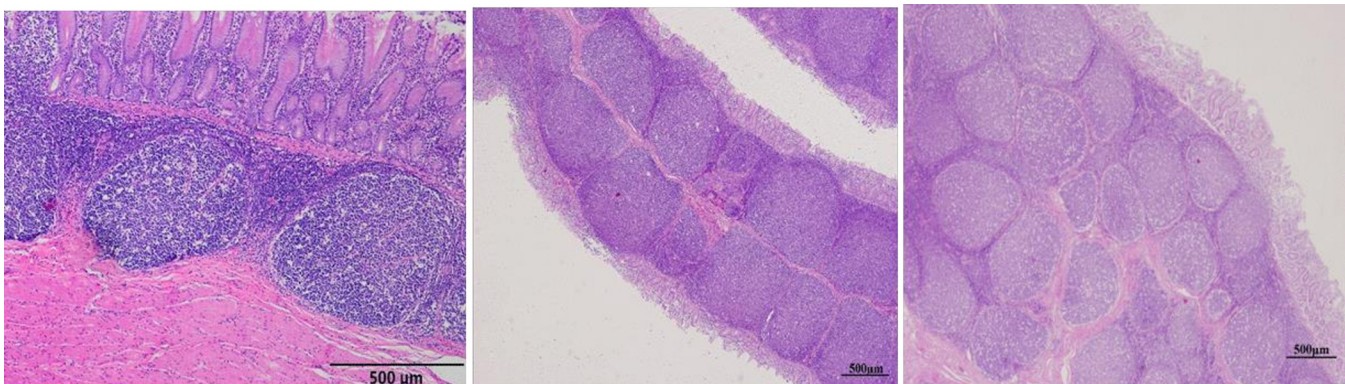

**Fig 8. Organization of the aggregated lymphoid nodules.** The lymphoid follicles were arranged in submucosa in a single row (A); within the fold in 2 rows (B); some follicles dispersed within the folds in more rows with multiple shapes and sizes (C). (H and E, 40x).

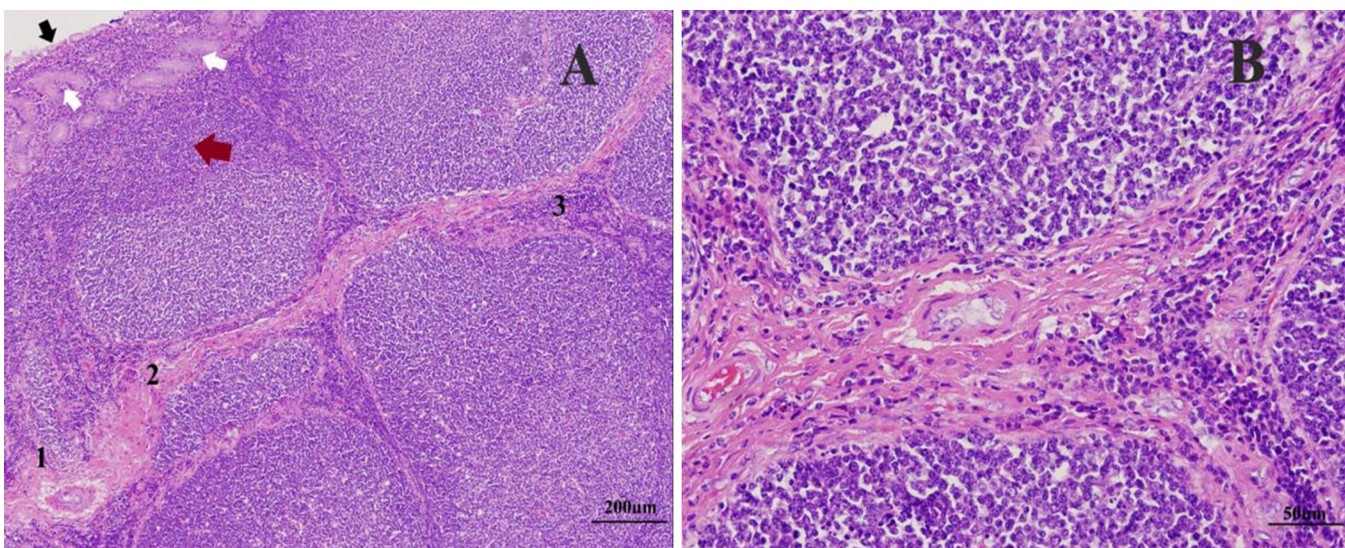

**Fig 9. Part of mucosal fold stained with (H and E).** (A): the epithelium (black arrows), mucous gland (white arrows), and solitary lymphoid tissue (red arrows) overlying the nodules with the interfollicular area containing HEV (1), connective tissue (2), and lymphoid cells (3). (B): high magnification of A thickening of the interfollicular area due to the infiltration of the connective tissue and lymphoid cells. (Original magnification: A = 100x, B = 400x).

### 3.6. Age-related differences in the IgA and IgG APCs distribution densities in the ALNA

In all groups, the analysis showed that the distribution densities of IgA+ APCs in the LP of the longitudinal area were all higher than the densities of the IgG+ APCs in the same site. Among these age groups, the differences between the densities of the IgA+ APCs were not significant in the longitudinal area of the young and old groups (except for 1.5 years and 2 years), but the densities of these cells in the longitudinal areas of the pubertal animals were significantly different from those of the young and old animals. Regarding the IgG+ APCs, there were significant differences between the densities of populations in the longitudinal areas of the young and old, as well as between pubertal and old groups. Also, between young and pubertal groups, significant differences were observed except for 8 months old animals compared to the pubertal group. Within the age group, there were no significant differences between the densities of the two types of APCs in the longitudinal area in all groups except in the young group (Fig 12).

## 4. Discussion

The stomach of the camel is not only morphologically distinct but also possesses unique immunological characteristics, such that it contains significant lymphoid tissue associated with the mucosa, which is considered an integral part of the mucosal immune system and the initial site of induction of mucosal immunity. According to the available literature, we are the first to identify and state an aggregation of lymphoid tissue in the stomach of the Dromedary camel. The stomach of the Dromedary camel consists of three compartments and third compartment is abomasum [28] The three compartments can be divided into the cardiac glandular area, the fundic glandular area and pyloric glandular area [29]. The current study is unique as it provides new information on the mucosa-associated lymphoid tissue in the stomach of Dromedary camels. The presence of aggregated lymphoid nodule area in the abomasum is a discovery in the camel stomach and sheds light on the immunological and physiological status of the camel stomach, making the stomach an essential immune organ. This discovery

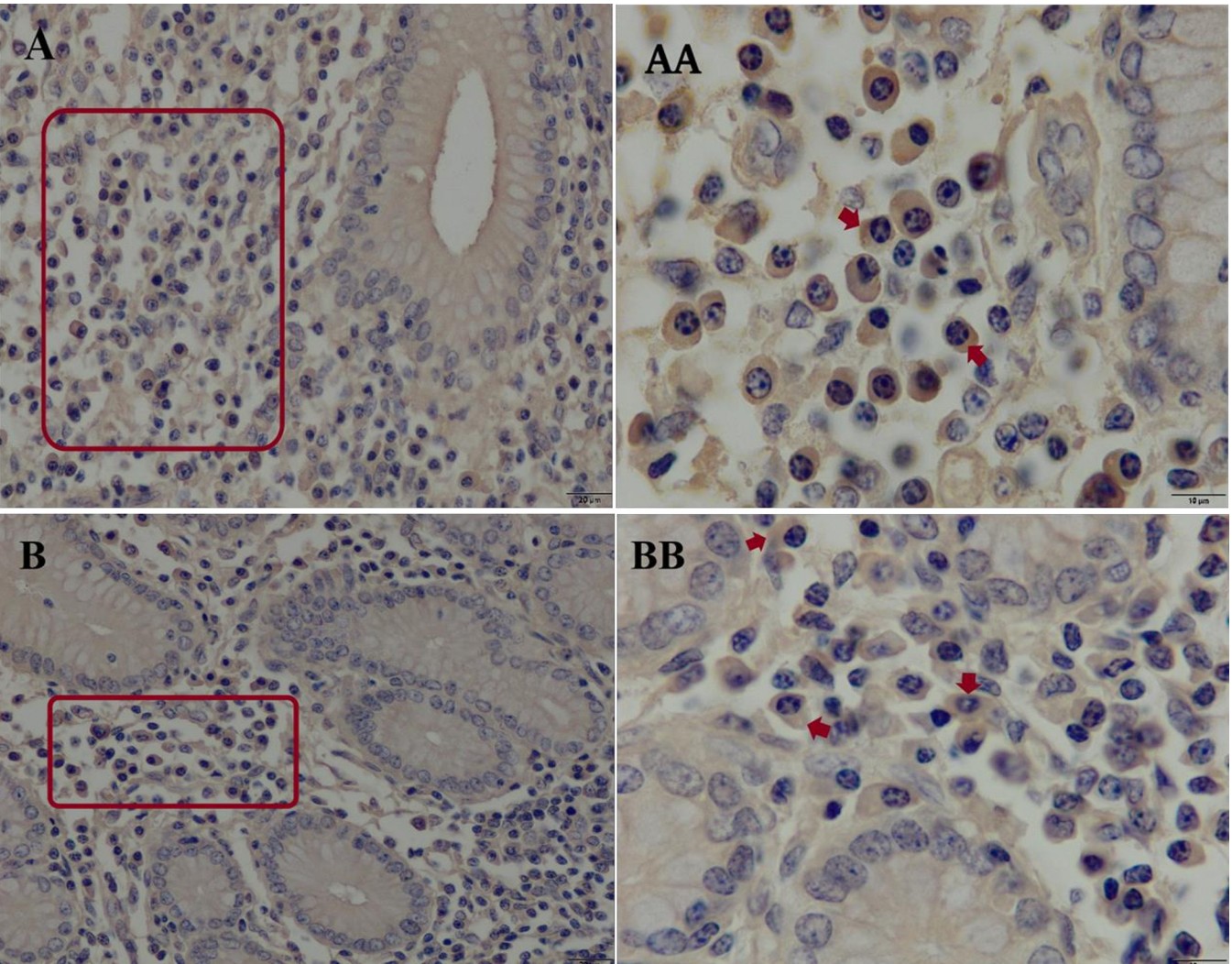

**Fig 10. Immunohistochemical staining of IgA antibody-producing cells in the ALNA of young Dromedary camels.** (A) Distribution of IgA APCs in longitudinal mucosal fold area; (B) Distribution of IgA APCs in reticular mucosal fold area. The IgA APCs were dispersed in the lamina propria among the gastric glands. Red rectangles are representative views from the two sublocations (Original magnification left column = 400×, right column = 1000×).

provided further immunomorphological evidence for the strong disease resistance of camels and also represented a new addition to the research field of mucosal immunology. Such results have never been reported in other animals or humans. The aggregated lymphoid follicles are an indispensable component of GALTs and are commonly found in the PPs of the small and large intestine and appendix in animals and humans. They are necessary for the healing of intestinal wounds by promoting the emigration and proliferation of epithelial cells in the wound-side crypts and reducing wound retraction [30]. The characteristics of aggregated lymph nodes (number, length, dissemination, and morphology) differ according to animal species, age, gut regions, and nutrition. Above the aggregated lymphoid nodules of the ileum in the submucosa, the mucosa has intestinal villi that together with the nodules elevate the surface into longitudinal folds in cattle and multiple fold orders in sheep [31–33]. While the ileal aggregated lymphoid in the Bactrian and Dromedary camel were mainly located in the submucosa and were devoid of intestinal glands. These findings were symmetrical to our observations

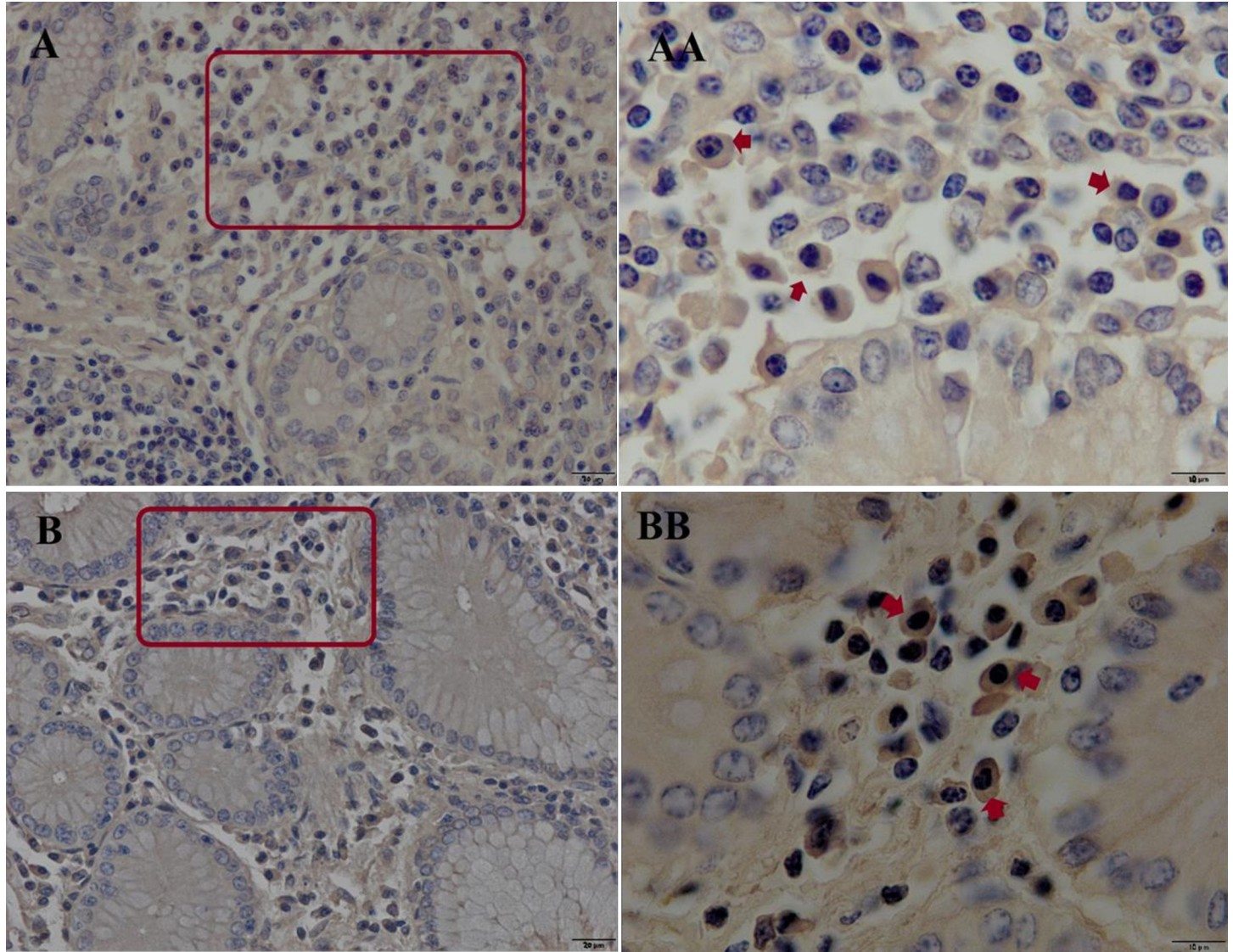

**Fig 11. Immunohistochemical staining of IgG antibody-producing cells in the ALNA of young Dromedary camels.** (A) Distribution of IgG APCs in longitudinal mucosal fold area; (B) Distribution of IgG APCs in reticular mucosal fold area. The IgG APCs were dispersed in the lamina propria among the gastric glands. Red rectangles are representative views from the two sublocations (Original magnification left column = 400×, right column = 1000×).

on the lymphoid nodes in the abomasum of Dromedary camel, whereas the ileal lymphoid nodes in the pig appeared as a band-like structure [34]. PPs are scattered throughout the small intestine in mice and contain 4–10 lymphoid nodules each, whereas the 30–80 aggregated lymphoid nodules in the human small intestine contain 5–900 lymphoid nodes each [21]. Analogous to the anatomical location and gross appearance of the lymphoid nodule region in the stomach of the dromedary and Bactrian camel, the lymphoid follicles are confined to a triangular zone on the ventral surface of the third compartment along the minor curvature. The region consists of a reticular and longitudinal membranous fold with a clear border between the region with the clustered lymphoid follicles stomach of the Dromedary camel consisted of aggregated lymphoid follicles lying in a row in the sub-mucosa and varied in size and shape. However, the follicles are arranged in multiple rows in certain mucosal folds, analogous to the

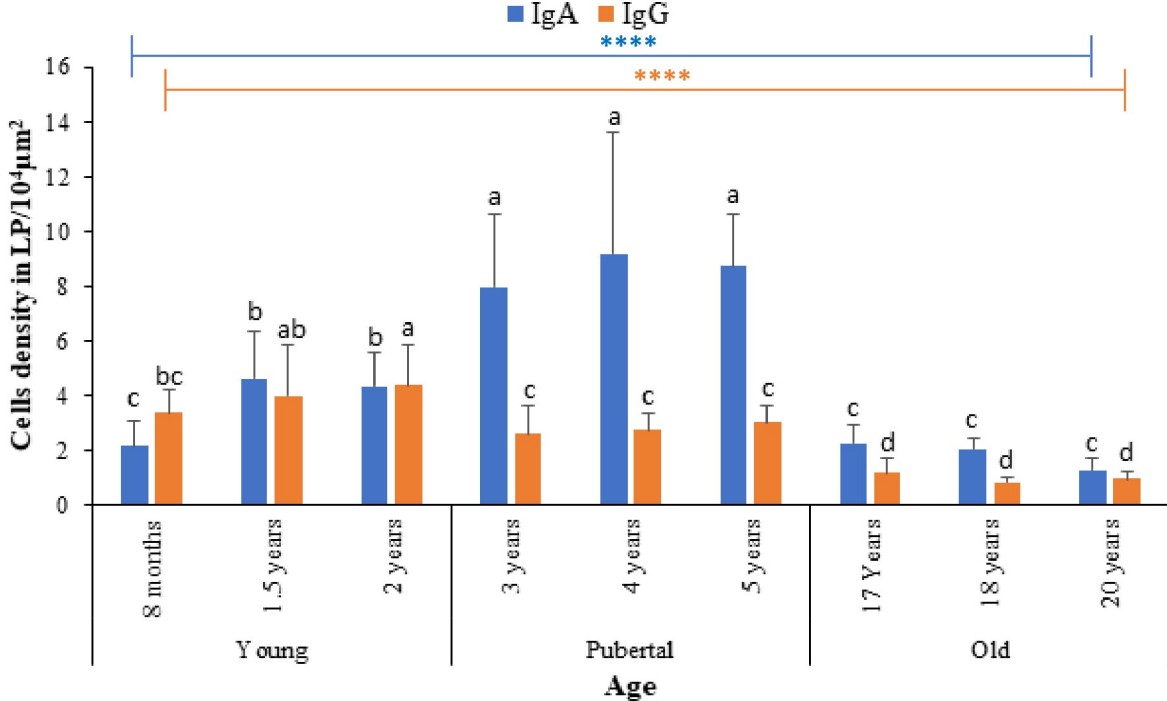

**Fig 12. Density distribution of IgA+ and IgG+ antibody-secreting cells in the ALNA of Dromedary camels of different ages.** Data are expressed as means (n = 3). Error bars represent standard deviation. Bars with a different letter are significantly different (one-way ANOVA with Duncan's multiple range test, p < 0.05). ****, represents significance at p < 0.0001.

ileal patches in Dromedary camels [35] and other animals. Some bottle-like lymphoid nodules penetrated the lamina propria, which overlapped with the follicle-associated epithelium (FAE). This is consistent with previous findings in the Bactrian camel stomach, where bottle-shaped nodules have been reported. High endothelial venules were found between the lymphoid follicles, as in the lymphoid nodules of the Bactrian camel stomach and the ileal nodules of the dromedary camel and other species [36]. The endothelial venules were the only entry point for lymphocyte recirculation. A similar result was found in the aggregated lymphoid nodules in the ileal PPs of Dromedary and Bacterian camels, sheep and cattle. In agreement with the results obtained in the study of the ileal lymphoid nodules in Dromedary camels, sheep, and cattle, poor development of the nodules and a decrease in the number and size of the clustered lymphoid follicles with age were observed in adult camels.

The observations in the current study show that IgA+ APCs were mainly distributed in the gastric lamina propria of the Abomasum's ALNA in Dromedary camels, and that most of the cells gathered among the gastric glands. Further, the distribution features of IgG+ APCs were similar to those of IgA+ APCs and these characteristics do not change with age. Although both of the IgA and IgG APCs were unequally distributed in LP, the statistical analysis showed that the densities of the IgA+ APCs were higher than those of IgG+ APCs in the longitudinal mucosal fold's area of the ALNA. The analysis also revealed that the distribution densities of the two APCs populations consistently increased from the young towards puberty peak in pubertal age and then decreased in old ages, which is consistent with the changes observed in the anatomical and histological properties of the stomach's lymphoid tissues.

These results could be further confirmation and explanation that the immunological significance of this structure changes with the aging of the animal.

## 5. Conclusion

In this study, anatomical, histological and immunohistochemical approaches of the abomasal mucosa were performed. The uniqueness of this study is that the lymphoid tissue was carefully examined in association with the Omasal mucosa, which has never been done before in Dromedary camels. Our results showed that the stomach of the single-humped camel is an important immune organ and adds new content to the fields of research in immunology. To date, there is no information on the presence of organized lymphoid nodules in the stomachs of other animals and this may be one of the camel's specialties. The function of this novel PPs-like structure in the stomach needs further study.

## Acknowledgments

We are grateful to professor Haider Ibrahim and Dr. Murtada mahgoub (College of Veterinary Medicine, University of Bahri, Sudan) for their support during the specimens and samples collection.

## Author Contributions

**Conceptualization:** Wen-Hui Wang.

**Data curation:** Zubieda Ibrahim Hassan Omer.

**Formal analysis:** Zubieda Ibrahim Hassan Omer.

**Funding acquisition:** Wen-Hui Wang.

**Investigation:** Zubieda Ibrahim Hassan Omer.

**Methodology:** Zubieda Ibrahim Hassan Omer, Jia Lu, Yu-Jiao Cheng.

**Resources:** Jia Lu.

**Software:** Pei-Xuan Li.

**Supervision:** Wen-Hui Wang.

**Visualization:** Zhi-Hua Chen.

**Writing – original draft:** Zubieda Ibrahim Hassan Omer.

**Writing – review & editing:** Zubieda Ibrahim Hassan Omer, Zhi-Hua Chen, Wen-Hui Wang.

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
