## [Decision Letter · Decision Letter 0]

7 Jun 2022

PONE-D-22-11222Discovery and observation on the anatomical and histological characteristics of the aggregated lymphoid nodules in the stomach of Dromedary camels (camelus Dromedarius)PLOS ONE

Dear Dr. Wang,

Thank you for submitting your manuscript to PLOS ONE. After careful consideration, we feel that it has merit but does not fully meet PLOS ONE’s publication criteria as it currently stands. Therefore, we invite you to submit a revised version of the manuscript that addresses the points raised during the review process.

The manuscript has been assessed by two experts in the field; in particular, I would advise you to pay attention to the comments from Reviewer 2. Please find the detailed critiques in the reviews. Please address in your response letter all of them, however, please pay a

particular attention to the followings:

1) Please re-consider the suggested title change as suggested by Reviewer 1.

2) Please address reviewer 2 concern about the redundant nature of the findings since a similar lymphoid organization has been reported in the abomasum of camels.

3) Please confirm the lymphoid nature of the nodules by molecular characterization such as the presence of lymphocytes in the tissue. 

4) Please carefully check and revise grammatical and typing errors in the whole manuscript.

5) Please address the statistical analysis discrepancy as suggested by Reviewer 1.

We look forward to receiving your revised manuscript.

Kind regards,

Nirakar Sahoo, PhD

Academic Editor

PLOS ONE

Journal Requirements:

Reviewers' comments:

Reviewer's Responses to Questions

**Comments to the Author**

1. Is the manuscript technically sound, and do the data support the conclusions?

Reviewer #1: Yes

Reviewer #2: Yes

2. Has the statistical analysis been performed appropriately and rigorously? 

Reviewer #1: No

Reviewer #2: Yes

3. Have the authors made all data underlying the findings in their manuscript fully available?

Reviewer #1: Yes

Reviewer #2: Yes

4. Is the manuscript presented in an intelligible fashion and written in standard English?

Reviewer #1: No

Reviewer #2: No

5. Review Comments to the Author

Reviewer #1: In the manuscript - ‘Discovery and observation on the anatomical and histological characteristics of the aggregated lymphoid nodules in the stomach of Dromedary camels (camelus Dromedarius)’, H-Ibrahim et al. have reported an aggregation of lymphoid tissues present in the stomach of the Dromedary camel, which thereby indicates that the stomach of Dromedary camels also serves as an essential immune organ. The work is novel as in the case of Dromedary camels. Authors have sufficiently screened the previously published pieces of literature in relevance to the work. The study is well organized and sufficiently discussed. The manuscript has enough potential to be considered for its publication; however, the current version of the manuscript needs to be improved. There are some minor issues in the current version, authors are suggested to address these points before its final acceptance:

1. The apt title of the study according to the presented results shown in the manuscript would be ‘Age-dependent changes in the anatomical and histological characteristics of the aggregated lymphoid nodules in the stomach of Dromedary camels’. The study of age-dependent changes of ALNA are very interesting.

2. Although the manuscript is readable, understandable, and scientifically sound, the authors are suggested to carefully check and revise grammatical and typing errors in the whole manuscript. For example, in line 30, it is written that Gastrointestinal associated lymphoid tissue (GALT) is important component of the mucosal immune system, while an article is missing in the sentence. It should be written as Gastrointestinal associated lymphoid tissue (GALT) is an important component of the mucosal immune system. In another example, in lines 50-52, it is written as ‘The histological characteristics of the structure was the same as PPs in the intestine of dromedary camel while its anatomical appearance showed some dissimilarity.’ It should be written like ‘The histological characteristics of the structure were the same as PPs in the intestine of the dromedary camel, while anatomical appearance showed some difference.’ The use of articles and commas is missing. In line 229, the full stop is missing. Please also check extra spaces and carefully revise the manuscript.

3. Why the authors have used unnecessary hyphen (-), e.g., in line 92: nod-ules, line 155: de-creasing, in line 159: Ja-pan, and at other places there are many more such types of hyphen used? Any reason?

4. In line 153, what is the fixed volume size of tissue block in mm3? Any number?

5.In fig. 2, please show a triangular zone either with the dotted line triangular zone or an arrow as according to the written sentences in lines 175-176.

6. In fig. 1-4, the calibration bar is missing. Please increase the font size of the calibration label in fig. 7-9.

7. In fig. 9 legends, magnification of microscope objective is written 400x. Is it true?

8. In line, 258, what does mean by HIV. Please clarify the abbreviation so that non-expert can understand.

9. In line 275, the term ‘department’ should be replaced with ‘compartment’.

10. In statistical analysis, fig. 5 and fig. 6, the authors did not show the significant relationships clearly. The authors are suggested to use a line bar along with streak(s) (*) at the top of graph to show significant relationships between experimental subjects.

Reviewer #2: In this manuscript entitled “Discovery and observation on the anatomical and histological characteristics of the aggregated lymphoid nodules in the stomach of Dromedary camels (Camelus Dromedarius)”, H-Ibrahim, et al. report the discovery of aggregated lymphoid nodules in the abomasum of Dromedary camels and therefore propose an immunological role of the stomach of this organism.

1) In the introduction, the authors mentioned that such lymphoid tissues had been previously identified in the abomasum of Bactrian camel apart from pigs. In my opinion, the same observation in another species makes the discovery redundant.

2) To confirm presence of the lymphoid nodules, the experiments undertaken by the authors are solely anatomical and histological in nature. However, no molecular characterization was carried out in order to confirm the lymphoid status, for example, presence of lymphocytes, etc. This makes me wonder whether this manuscript falls under the purview of Plos One, or is more appropriate for a journal publishing anatomical/histological characterizations.

6. PLOS authors have the option to publish the peer review history of their article (what does this mean?). If published, this will include your full peer review and any attached files.

Reviewer #1: **Yes: **Dr. Navin Kumar Ojha

Reviewer #2: No

---

## [Author Response · Author response to Decision Letter 0]

22 Aug 2022

Comment.1

Has the statistical analysis been performed appropriately and rigorously?

Reviewer #1: No

Reviewer #2: Yes

Response: Thank you for this suggestion, we appreciate your comment. We have changed the way we analyzed the data please see figure 5 and 6.

Comment.2

Is the manuscript presented in an intelligible fashion and written in standard English? Reviewer #1: No

Reviewer #2: No

Response: We are grateful for your comment. We have revised the language and corrected all grammatical and typing error. The manuscript has been revised by a native English speaker.

Comment.3

The apt title of the study according to the presented results shown in the manuscript would be ‘Age-dependent changes in the anatomical and histological characteristics of the aggregated lymphoid nodules in the stomach of Dromedary camels. The study of age-dependent changes of ALNA are very interesting.

Response: Agreed. Thank you for this valuable comment, we have changed the title “Age-dependent changes in the anatomical and histological characteristics of the aggregated lymphoid nodules in the stomach of Dromedary camels (Camelus Dromedarius)”

Comment.4

Although the manuscript is readable, understandable, and scientifically sound, the authors are suggested to carefully check and revise grammatical and typing errors in the whole manuscript. For example, in line 30, it is written that Gastrointestinal associated lymphoid tissue (GALT) is important component of the mucosal immune system, while an article is missing in the sentence. It should be written as Gastrointestinal associated lymphoid tissue (GALT) is an important component of the mucosal immune system. In another example, in lines 50-52, it is written as ‘The histological characteristics of the structure was the same as PPs in the intestine of dromedary camel while its anatomical appearance showed some dissimilarity.’ It should be written like ‘The histological characteristics of the structure were the same as PPs in the intestine of the dromedary camel, while anatomical appearance showed some difference.’ The use of articles and commas is missing. In line 229, the full stop is missing. Please also check extra spaces and carefully revise the manuscript.

 Response: thank you very much for these comments. we have carefully checked and corrected this part please see line 28 and line (48~50). We also checked extra spaces.

Comment.5

Why the authors have used unnecessary hyphen (-), e.g., in line 92: nod-ules, line 155: de-creasing, in line 159: Ja-pan, and at other places there are many more such types of hyphens used? Any reason?

Response: Thank you very much for these important notices. We have corrected these mistakes.

Comment.6

In line 153, what is the fixed volume size of tissue block in mm3? Any number?

Response: Thank you, we have added the volume size of tissue block (1mm3). Please see line (150).

Comment.7

In fig. 2, please show a triangular zone either with the dotted line triangular zone or an arrow as according to the written sentences in lines 175-176.

Response: Thank you, we have shown the triangular zone with arrows. please see figure 2.

Comment.8

In fig. 1-4, the calibration bar is missing. Please increase the font size of the calibration label in fig. 7-9.

Response: Agreed. We have increased the font size of the calibration label in fig. 7-9.

Comment.9

In fig. 9 legends, magnification of microscope objective is written 400x. Is it true?

Response: the magnification of the microscopic objective is 400x, we observed it in the microscope using the lens 40.

Comment.10

In line, 258, what does mean by HIV. Please clarify the abbreviation so that non-expert can understand.

Response: Thank you for this important comment. We have corrected the mistake. Please see line 255

Comment.11

In line 275, the term ‘department’ should be replaced with ‘compartment’.

Response: Agreed. Please See line 273.

Comment.12

 In statistical analysis, fig. 5 and fig. 6, the authors did not show the significant relationships clearly. The authors are suggested to use a line bar along with streak(s) (*) at the top of graph to show significant relationships between experimental subjects.

Response: thank you very much for much for this valuable comments. We have changed the way we analyzed the data. Please see figure 5 and 6.

Comment.13

In the introduction, the authors mentioned that such lymphoid tissues had been previously identified in the abomasum of Bactrian camel apart from pigs. In my opinion, the same observation in another species makes the discovery redundant.

Response: Agreed. We have removed this part from the introduction.

Comment.14

 To confirm presence of the lymphoid nodules, the experiments undertaken by the authors are solely anatomical and histological in nature. However, no molecular characterization was carried out in order to confirm the lymphoid status, for example, presence of lymphocytes, etc. This makes me wonder whether this manuscript falls under the purview of Plos One, or is more appropriate for a journal publishing anatomical/histological characterizations.

Response: We are grateful for your suggestion. we have done the immunohistochemistry and detected the immunoglobulin–producing cells please see the results (section 3.5).

---

## [Decision Letter · Decision Letter 1]

21 Oct 2022

PONE-D-22-11222R1Discovery and observation on the anatomical and histological characteristics of the aggregated lymphoid nodules in the stomach of Dromedary camels (camelus Dromedarius)PLOS ONE

Dear Dr. Wang,

Thank you for submitting your manuscript to PLOS ONE. After careful consideration, we feel that it has merit but does not fully meet PLOS ONE’s publication criteria as it currently stands. Therefore, we invite you to submit a revised version of the manuscript that addresses the points raised during the review process.

The manuscript has been assessed by two experts in the field; please find their comments appended at the end of this email. I would advise you to pay attention to  Reviewer 1.

We look forward to receiving your revised manuscript.

Kind regards,

Nirakar Sahoo, PhD

Academic Editor

PLOS ONE

Journal Requirements:

Reviewers' comments:

Reviewer's Responses to Questions

**Comments to the Author**

1. If the authors have adequately addressed your comments raised in a previous round of review and you feel that this manuscript is now acceptable for publication, you may indicate that here to bypass the “Comments to the Author” section, enter your conflict of interest statement in the “Confidential to Editor” section, and submit your "Accept" recommendation.

Reviewer #1: (No Response)

Reviewer #2: All comments have been addressed

2. Is the manuscript technically sound, and do the data support the conclusions?

Reviewer #1: Partly

Reviewer #2: Yes

3. Has the statistical analysis been performed appropriately and rigorously? 

Reviewer #1: No

Reviewer #2: Yes

4. Have the authors made all data underlying the findings in their manuscript fully available?

Reviewer #1: Yes

Reviewer #2: Yes

5. Is the manuscript presented in an intelligible fashion and written in standard English?

Reviewer #1: No

Reviewer #2: Yes

6. Review Comments to the Author

Reviewer #1: The revised manuscript is improved, however, its presentation quality still does not match the standard of the journal. Although scientific data are organized and conclusions are in the line of results, without rigorous and clear statistical analysis, results are difficult to conclude. Additionally, the representation of results is not consistent. Therefore, the authors are strongly suggested to revise the manuscript very carefully before its final decision to be accepted for publication.

Major issue:

1. The authors have mentioned that the statistically significant differences in the width and height of the reticular and longitudinal mucosal folds of the abomasum ALNA between groups; were determined by one-way analysis of variance (ANOVA). However, statistical significance is not clearly described either in the text or in the figure legends. Based on p-values, no streak (*) can be seen in any figure.

2. In lines 331 – 333, it is mentioned- ‘However, there were significant differences……….but there was no significant difference between young and pubertal groups (P < 0.0001).’ This indicates either the p-value is written at the wrong place or the interpretation of data is wrong.

Authors are suggested to revise the manuscript carefully in terms of statistical analysis and their outcome in texts.

Minor issues:

1. Fig. 1 – 4 have no calibration bar.

2. In line, the volume of a block should be written as 1 cm3 instead of 1 cm. Please write an appropriate unit.

3. Fig. 7 and 8 –the calibration bar and its labeling are not consistent for all three figures.

4. In fig. 5 and 6, what is the meaning of alphabets- a, b, c…….. Please use streak to represent significance.

5. Fig. 10 and 11 – Difficult to see calibration bar. Please describe red rectangles in the figure legend.

Reviewer #2: To address the concerns of this reviewer, the authors have performed immunohistochemistry experiments to confirm the lymphoid status of the nodules. I have no further concerns and recommend this manuscript for publication.

7. PLOS authors have the option to publish the peer review history of their article (what does this mean?). If published, this will include your full peer review and any attached files.

Reviewer #1: **Yes: **Dr. Navin Kumar Ojha

Reviewer #2: No

---

## [Author Response · Author response to Decision Letter 1]

23 Nov 2022

Comment.1

Has the statistical analysis been performed appropriately and rigorously?

Reviewer #1: No

Reviewer #2: Yes

Response: Thank you, We have addressed Reviewer 1 comment, please see the statistical results and figures.

Comment.2

Is the manuscript presented in an intelligible fashion and written in standard English? Reviewer #1: No

Reviewer #2: Yes

Response: We are grateful for your comment. We have revised the language. The manuscript has been revised by a native English speaker.

Review Comments to the Author

Reviewer #1: The revised manuscript is improved; however, its presentation quality still does not match the standard of the journal. Although scientific data are organized and conclusions are in the line of results, without rigorous and clear statistical analysis, results are difficult to conclude. Additionally, the representation of results is not consistent. Therefore, the authors are strongly suggested to revise the manuscript very carefully before its final decision to be accepted for publication.

Major issue:

1. The authors have mentioned that the statistically significant differences in the width and height of the reticular and longitudinal mucosal folds of the abomasum ALNA between groups; were determined by one-way analysis of variance (ANOVA). However, statistical significance is not clearly described either in the text or in the figure legends. Based on p-values, no streak (*) can be seen in any figure.

Response: Agreed. Thank you for this valuable comment, we have described the statistical significance in both the text and in the figure’s legends. Streaks have been shown. Please see the figures.

2. In lines 331– 333, it is mentioned- ‘However, there were significant differences………. but there was no significant difference between young and pubertal groups (P < 0.0001).’ This indicates either the p-value is written at the wrong place or the interpretation of data is wrong.

Response: Thank you for drawing our attention to this . this part has been addressed accordingly.

Minor issues:

1. Fig. 1 – 4 have no calibration bar.

Response: Thank you very much for this comment. We would like to inform that these figures has been taken from the field by normal camera.

2. In line, the volume of a block should be written as 1 cm3 instead of 1 cm. Please write an appropriate unit.

Response: please the correction has been made in the manuscript at line 144.

3. Fig. 7 and 8 – the calibration bar and its labeling are not consistent for all three figures.

Response: Thank you for drawing our attention again to this important point. This part has been addressed. please See line (272 – 276) and (282 – 285). 

4. In fig. 5 and 6, what is the meaning of alphabets- a, b, c…….. Please use streak to represent significance.

Response: Bars with a different letter are significantly different. Streaks has been shown please see the figures.

5. Fig. 10 and 11 – Difficult to see calibration bar. Please describe red rectangles in the figure legend.

Response: Thank you. This time our figures are clear. Red rectangle has been described.

Reviewer #2: To address the concerns of this reviewer, the authors have performed immunohistochemistry experiments to confirm the lymphoid status of the nodules. I have no further concerns and recommend this manuscript for publication.

Response: We are grateful to you . Thanks for the all reviewer's for their time and comments. We hope our manuscript would finally meet all reviewer’s favorable consideration and approval to be published in this highly respected journal.

---

## [Editor Report · Decision Letter 2]

6 Dec 2022

Discovery and observation on the anatomical and histological characteristics of the aggregated lymphoid nodules in the stomach of Dromedary camels (camelus Dromedarius)

PONE-D-22-11222R2

Dear Dr. Wang,

We’re pleased to inform you that your manuscript has been judged scientifically suitable for publication and will be formally accepted for publication once it meets all outstanding technical requirements.

Kind regards,

Nirakar Sahoo, PhD

Academic Editor

PLOS ONE
---

## [Editor Report · Acceptance letter]

9 Jan 2023

PONE-D-22-11222R2 

Age-dependent changes in the anatomical and histological characteristics of the aggregated lymphoid nodules in the stomach of Dromedary camels (Camelus Dromedarius) 

Dear Dr. Wang:

I'm pleased to inform you that your manuscript has been deemed suitable for publication in PLOS ONE. Congratulations! Your manuscript is now with our production department. 

Kind regards, 

on behalf of

Dr. Nirakar Sahoo 

Academic Editor

PLOS ONE